# A Rational Designed Novel Bispecific Antibody for the Treatment of GBM

**DOI:** 10.3390/biomedicines9060640

**Published:** 2021-06-03

**Authors:** Rui Sun, Yuexian Zhou, Lei Han, Zhidi Pan, Jie Chen, Huifang Zong, Yanlin Bian, Hua Jiang, Baohong Zhang, Jianwei Zhu

**Affiliations:** 1Engineering Research Center of Cell & Therapeutic Antibody, Ministry of Education, School of Pharmacy, Shanghai Jiao Tong University, 800 Dongchuan Road, Minhang District, Shanghai 200240, China; sun-rui@sjtu.edu.cn (R.S.); yuexianzhou@sjtu.edu.cn (Y.Z.); panzhidi@sjtu.edu.cn (Z.P.); jacy0214@sjtu.edu.cn (J.C.); zhaoxiliunian@sjtu.edu.cn (H.Z.); weixiaobyl@sjtu.edu.cn (Y.B.); 2Jecho Biopharmaceuticals Co., Ltd. No. 2018 Zhongtian Avenue, Binhai New Area, Tianjin 300457, China; lei_han@alumni.sjtu.edu.cn (L.H.); hjiang@jechoinc.com (H.J.); 3Jecho Biopharmaceutical Institute, No. 58 Yuanmei Road, Minhang District, Shanghai 200241, China; 4Jecho Laboratories Inc., 7320 Executive Way, Frederick, MD 21704, USA

**Keywords:** bispecific antibody, GBM, EGFRvIII, split intein

## Abstract

Epidermal growth factor receptor variant III (EGFRvIII) is highly and specifically expressed in a subset of lethal glioblastoma (GBM), making the receptor a unique therapeutic target for GBM. Recently, bispecific antibodies (BsAbs) have shown exciting clinical benefits in cancer immunotherapy. Here, we report remarkable results for GBM treatment with a BsAb constructed by the “BAPTS” method. The BsAb was characterized through LC/MS, SEC-HPLC, and SPR. Furthermore, the BsAb was evaluated in vitro for bioactivities through FACS, antigen-dependent T-cell-mediated cytotoxicity, and a cytokine secretion assay, as well as in vivo for antitumor activity and pharmacokinetic (PK) parameters through immunodeficient NOD/SCID and BALB/c mouse models. The results indicated that the EGFRvIII-BsAb eliminated EGFRvIII-positive GBM cells by recruiting and stimulating effector T cells secreting cytotoxic cytokines that killed GBM cells in vitro. The results demonstrated the antitumor potential and long circulation time of EGFRvIII-BsAb in NOD/SCID mice bearing de2–7 subcutaneously heterotopic transplantation tumors and BALB/c mice. In conclusion, our experiments in both in vitro and in vivo have shown the remarkable antitumor activities of EGFRvIII-BsAb, highlighting its potential in clinical applications for the treatment of GBM. Additional merits, including a long circulation time and low immunogenicity, have also made the novel BsAb a promising therapeutic candidate.

## 1. Introduction

GBM is a highly malignant tumor that originates in the central nervous system (CNS). It accounts for 54% of all gliomas and 16% of all primary brain tumors [1]. For newly diagnosed patients, the five-year survival rate is only 6.8% [2]. Over the past 30 years, little improvement has been made in treating GBM. Possible reasons include a highly heterogeneous brain tumor microenvironment, an impermeable blood–brain barrier (BBB), and a lack of T-cell infiltration. GBM is characterized by rapid proliferation, invasiveness, and poor prognosis [3]. The current standard of care is still external irradiation after maximum safe resection, treatment with temozolomide (TMZ), and maintenance chemotherapy [4,5].

EGFRvIII is an EGFR mutant, an 801 bp in-frame deletion spanning exons 2–7 of the coding sequence [6,7,8], and is highly and specifically expressed in a subset of human GBM. The EGFRvIII loses its binding site to the ligand EGF, so it exhibits a low level of constitutive activity due to impaired endocytosis and degradation [9]. Furthermore, cells with the EGFRvIII confer significant tumor growth advantages and resistance to chemotherapy and radiation therapy [10,11]. The mutant EGFRvIII has been detected in various cancers, including brain, breast, ovarian, lung, and prostate cancers, but not in normal tissue, which makes it a potential tumor-specific antigen (TSA) for cancer therapy.

Bispecific antibodies (BsAbs) have shown excellent potential in the treatment of cancers. Until recently, there were four BsAbs approved on the market—Removab [12], Blincyto [13,14], Hemlibra [15], and Rybrevant (EP2922872A1)—and many others in clinical trials [16,17]. The T-cell-engaged bispecific antibody (TCB) that recruits and activates circulating T cells to tumor sites has attracted extensive research attention. Here, we would like to develop an IgG format of EGFRvIII-BsAb, a TCB antibody, targeting CD3 and EGFRvIII. We expected that a BsAb with an IgG format would have low immunogenicity and a prolonged body circulation time. Until recently, chain-mispairing, especially light chain-mispairing, has been a major issue in the generation of a BsAb. Thus, a “Bispecific Antibody by Protein Trans-splicing” (BAPTS) platform, developed in our lab [18,19,20], was employed to synthesize the EGFRvIII-BsAb for this study.

Since the uncontrolled cytokine release of an Fc-equipped BsAb is due to CD3 aggregates on T cells that may not be conducive to antitumor effects [16,21], we designed the EGFRvIII-specific BsAb with attenuated antibody-dependent cellular cytotoxicity (ADCC) and complement-dependent cytotoxicity (CDC) functions, targeting EGFRvIII-expressing GBM by recruiting T cells. This design may validate the suggestion that BsAbs with an IgG format have an extended half-life and durable tumor inhibition potential, as previously reported [22,23].

## 2. Materials and Methods

### 2.1. Animal and Tumor Cell Lines

Female NOD/SCID mice and male BALB/c mice were purchased from Charles River Laboratories in China and fed according to guidelines from the Institutional Animal Care and Use Committee of the School of Pharmacy of Shanghai Jiao Tong University (SJTU). The U87MG.ΔEGFR cell line was a gift from Renji Hospital, affiliated with the School of Medicine at SJTU. U87MG and Jurkat cells were purchased from the Chinese Type Culture Collection and preserved in our laboratory. All cell lines were cultured under standard conditions and used within 6 months after resuscitation without reauthentication. The HEK293E cell line and CHO-S cell line were preserved in our lab.

### 2.2. Protein Expression and Purification

The ADCC and CDC functions were attenuated by introducing mutations at L234A, L235A, and P329G (LALA-PG) in the Fc region of the EGFRvIII-BsAb, the EGFRvIII mAb, and the CD3 mAb, respectively. The CD3 protein fragment (fragment A in the BAPTS platform) was expressed by a stable transfected CHO cell line. Meanwhile, the EGFRvIII protein fragment (fragment B in the BAPTS platform) was transiently expressed by HEK293E cells, as previous reported [24,25]. Both fragment A and fragment B were captured by affinity chromatography with Capto L (GE Healthcare, Chicago, IL, USA). The EGFRvIII-BsAb was synthesized by the BAPTS method [20]. The complete BsAb was purified via isometric-precipitation and subsequent MMC ImpRes Multimodal Chromatography Column (Cytiva, Marlborough, MA, USA). Likewise, both controls, EGFRvIII mAb and CD3 mAb, were transiently expressed by HEK293E cells and purified with Protein A (GE Healthcare, Chicago, IL, USA) affinity chromatography. The EGFRvIII mAb amino acid sequence was identical to that of the 806 mAb (CN102405235A). All the recombinant antibodies were dialyzed overnight into phosphate buffer saline (PBS) and sterilized by 0.22 μM filtration.

### 2.3. Optimization of the Condition of the “BAPTS” Reaction

After being captured and purified via standard Protein L affinity chromatography, fragment A and fragment B were fused to form the BsAb by the trans-splicing reaction. In the absence or presence of reducing agents DTT (10708984001; Sigma-Aldrich, St. Louis, MO, USA) or TCEP (C4706-2G, Sigma-Aldrich, St. Louis, MO, USA) at various concentrations (0 mM, 0.01 mM, 0.05 mM, 0.1 mM, 0.5 mM, 1 mM, 2 mM, and 5 mM), the trans-splicing reaction occurred through the two fragments at a 1 mg/mL concentration and 37 °C for 2 h. SDS-PAGE was used to monitor the reaction progress. Upon the completion of the trans-splicing reaction, the product was dialyzed into PBS buffer, followed by an oxidization reaction in the presence of DHAA (D8132; Sigma-Aldrich, St. Louis, MO, USA) in a 20-molar excess (eq.) over TCEP at 37 °C for 3 h. The process of the oxidization reaction was also monitored by SDS-PAGE analysis.

### 2.4. EGFRvIII-BsAb Characterization

The purified EGFRvIII-BsAb was characterized with LC/MS, SEC-HPLC, and other methods. The molecular weight of the produced EGFRvIII-BsAb was determined with an LC/MS method. A sample of 0.5–1 mg was concentrated to 10 mg/mL and dialyzed into N-glycan excision buffer (5 mM NH_4_HCO_3_, 40867; Fluka, Charlotte, N.C., USA) and mixed with 1 μL PNGase F (500 U/μL, P0704S; New England Biolab, Ipswich, MA, USA) at 37 ℃ for 24 h to remove the N-glycan. Then, the components in the sample were separated with the ACQUITY UPLC Protein BEH C4 Column (186004495; Waters, Etten-Leur, Netherlands). A Waters Acquity VION IMS Q-Tof mass spectrometer (Milford, MA, USA) was coupled with the UPLC (Milford, MA, USA) to determine the mass of the target protein. The data were collected and processed with UNIFI 1.8 software (Waters, Milford, MA, USA). Purity of the EGFRvIII-BsAb was determined by the SEC-HPLC method (Agilent, Santa Clara, CA, USA).

### 2.5. Flow Cytometry

Flow cytometry was used to evaluate the EGFRvIII antigen expression level of GBM cell lines, the binding of the EGFRvIII-BsAb with the target tumor cell lines, and the binding of the EGFRvIII-BsAb to CD3+ Jurkat cells. An operating procedure following the manufacturer’s protocol (RRID AB_465926 and RRID AB_2565789) was followed to achieve the best results.

### 2.6. Binding of EGFRvIII-BsAb to EGFRvIII+ U87MG.ΔEGFR Cells

The GBM cell lines U87MG and U87MG.ΔEGFR, which were in a logarithmic growth phase, were harvested and incubated with EGFRvIII mAb and EGFRvIII-BsAb at a 1 ug/mL concentration on ice for 30 min and then washed twice with FACS buffer (PBS + 2% FBS) for the analysis of antigen expression. For the binding affinity analysis of the EGFRvIII-BsAb to U87MG.ΔEGFR and U87MG, cells were prepared with the same procedure, but with different antibody concentrations. Then, the PE-conjugated goat antihuman IgG-Fc antibody (12-4998-82; eBioscience^TM^, San Diego, CA, USA) was added and incubated on ice for 30 min. After being washed twice, samples were resuspended with FACS buffer, followed by analysis with a CytoFLEX cytometer (BECKMAN COULTER, Brea, CA, USA).

### 2.7. Binding of EGFRvIII-BsAb to CD3+ Jurkat Cells

Jurkat cells were resuspended in FACS buffer on ice and incubated with the CD3 mAb or the EGFRvIII-BsAb at a series of concentrations, followed by APC-conjugated antihuman IgG-Fc antibody (409305; Biolegend, San Diego, CA, USA). Samples were analyzed with a CytoFLEX cytometer using CytExpert software (BECKMAN COULTER, Brea, CA, USA).

### 2.8. Affinity Measurement of EGFRvIII-BsAb with Surface Plasmon Resonance (SPR)

The binding affinities of the antibodies were determined using SPR (Biacore 8K; GE Healthcare, Chicago, IL, USA). The EGFRvIII antigen (AVI10494; R&D System, Minneapolis, MN, USA) and extracellular domain of human CD3D/CD3E heterodimer (CT038-H2508H; Sino Biological, Beijing, China) were immobilized to a CM5 chip (29149603; GE Healthcare, Chicago, IL, USA) surface using standard protocols with 1-ethyl-3 (3-dimethylaminopropyl) carbodiimide (EDC)/N-hydroxysuccinimide (NHS) amine. The concentration series were fit to a 1:1 binding model to determine the equilibrium dissociation constant (KD) and association (Ka) and dissociation (Kd) rate constants. Surfaces were regenerated using injections of 0.1 M glycine (pH 1.5). EGFRvIII mAb and CD3 mAb were used as controls.

### 2.9. CD3+ Jurkat Cells Recruited to EGFRvIII+ Tumor Cells

EGFRvIII-positive U87MG.ΔEGFR cells were labeled with CFSE (65-0850-84; Invitrogen, Carlsbad, CA, USA) and CD3-positive Jurkat cells were labeled with PKH26 (PKH26GL; Sigma-Aldrich, St. Louis, MO, USA), according to the manufacturers’ protocols. The stained cells were mixed at a ratio of 1:1 and incubated with EGFRvIII-BsAb on ice for 1 h. EGFRvIII mAb was added as control. After the samples were washed twice and resuspended with FACS buffer, CFSE/PKH26 signals were analyzed with a CytoFLEX cytometer using Cyto-Expert software.

### 2.10. T-Cell Activation and Cytotoxic Cytokine Release in the Presence of Tumor Cells

Fresh human PBMCs were separated from healthy donors with Ficoll-Paque Plus density gradient media (17144003; GE Healthcare, Chicago, IL, USA) according to the manufacturer’s protocol, and then were incubated with 1 ug/mL EGFRvIII-BsAb in the presence of U87MG.ΔEGFR (EGFRvIII-positive cells) or U87MG (EGFRvIII-negative cells), respectively, with an effector-to-target (E/T) ratio of 10:1 for 28 h. Then, early signs of T-cell activation (CD69) were detected by FACS within CD4+ and CD8+ T-cell subsets. Cells were harvested and analyzed for T-cell activation with antihuman CD4 (10400-M001-P; Sino Biological, Beijing, China), antihuman CD8 (10980-MM28-F; Sino Biological, Beijing, China), and antihuman CD69 antibodies (560967; BD Biosciences, Franklin Lakes, NJ, USA). The culture supernatants at different incubation time points were also collected to detect the secreted level of IFN-γ (DY285; R&D System, Minneapolis, MN, USA) and IL-2 (DY202; R&D System, Minneapolis, MN, USA) by ELISA, according to the manufacturer’s protocol.

### 2.11. EGFRvIII-BsAb Mediated Cytotoxicity against GBM Cells

Target cells U87MG.ΔEGFR and control cells U87MG were seeded on a 96-well cell culture plate. After the cells adhered to the bottom of the plate, antibodies at different concentrations were preincubated at 37  °C in cell culture medium (no phenol RPMI 1640  +  10% FBS) for 30 min before adding the human PBMCs at an E/T ratio of 5:1. After coincubation for 75 h, the cell culture supernatant was collected, and the cytotoxicity was mediated by the EGFRvIII-BsAb or its parental EGFRvIII mAb, and the CD3 mAb was quantified via a CytoTox 96^®^ Non-Radioactive Cytotoxicity Assay Kit (G1780; Promega, Madison, WI, USA), following the standard procedure. All measurements were performed in triplicate. The percentage of cytotoxicity was calculated as follows:(1)$$\%\mathrm{cytotoxicity}=\frac{\mathrm{experimental}\;\mathrm{lysis}-\mathrm{spontaneous}\;\mathrm{effector}\;\mathrm{lysis}\;-\;\mathrm{spontaneous}\;\mathrm{target}\;\mathrm{lysis}}{\mathrm{maximum}\;\mathrm{target}\;\mathrm{lysis}\;-\;\mathrm{spontaneous}\;\mathrm{target}\;\mathrm{lysis}}\times \;100$$

### 2.12. Pharmacokinetics Analysis

Ten SPF-grade BALB/c male mice aged 8 weeks and weighing 20–26 g were selected and randomly divided into two groups to ensure that mouse status was basically the same between the two groups. The EGFRvIII-BsAb and the control antibody EGFRvIII mAb were single-intraperitoneally (i.p.) injected into mice at 5 mg/kg. The day of injection was d 0, and the blood samples were collected from the orbit at different time points: 15 min, 6 h, 1 d, 2 d, 4 d, 7 d, 10 d, 15 d, 21 d, and 28 d. The antibody concentration of serum samples was quantified by ELISA and PK parameters were determined with a noncompartmental analysis model using WinNonlin.

### 2.13. Xenograft Studies

To verify the antitumor effect of the EGFRvIII-BsAb and the existence of huPBMCs, female NOD/SCID mice aged 8 weeks (3/group) were subcutaneously implanted with the mixture of 3 × 10^6^ U87MG.ΔEGFR cells and 6 × 10^6^ or 3 × 10^6^ unstimulated human PBMCs (E/T 1:1 and E/T 2:1) on day 0. Treatments with the EGFRvIII-BsAb at 5 mg/kg and PBS started from the second day after inoculation, with 6 consecutive administrations to the mice every 3 days by intraperitoneal injection. To evaluate the dose-dependent prevention efficacy of the EGFRvIII-BsAb, another in vivo experiment was performed. Female NOD/SCID mice aged 8 weeks (5/group) were subcutaneously implanted with the mixture of 2.25 × 10^6^ U87MG.ΔEGFR cells and 3 × 10^6^ unstimulated human PBMCs on day 0. In order to compare with the control of EGFRvIII mAb on antitumor efficacy, EGFRvIII mAb at 1.5 mg/kg was also administrated. The drugs or controls (3 mg/kg EGFRvIII-BsAb, 1 mg/kg EGFRvIII-BsAb, 0.33 mg/kg EGFRvIII-BsAb, 1.5 mg/kg EGFRvIII mAb, or PBS) were administered from the same day of the inoculation, and 8 consecutive treatments were administered to the mice every three days by intraperitoneal injection. The tumor volume was detected every three days with a vernier caliper and calculated using the formula below:(2)$$\mathrm{approximated}\;\mathrm{formula}\;V=\frac{\;\mathrm{length}\;\times \;\mathrm{width}\;\times \;\mathrm{width}}{2}$$

After the mice were sacrificed, photographs of the stripped tumors were taken.

### 2.14. Animal Experiment Statement

All methods were conducted in accordance with guidelines from the Institutional Animal Care and Use Committee of the School of Pharmacy of SJTU, and all experimental protocols were approved by the Institutional Animal Care and Use Committee of the School of Pharmacy of SJTU. PBMCs were obtained from donators who signed an informed consent.

### 2.15. Statistical Analysis

Differences between samples indicated in the figures were tested for statistical significance by the Student’s *t*-test, and *p* < 0.05 was considered statistically significant.

## 3. Results

### 3.1. Preparation and Purification of the EGFRvIII-BsAb with the BAPTS Platform

We adapted the BAPTS platform to generate an IgG-like bispecific antibody targeting EGFRvIII and CD3 with minimum light/heavy chain mispairings, as reported by our lab [18,20]. In this method, two antibody fragments were expressed and purified, followed by the autocatalytic “protein trans-splicing” (PTS) reaction to conjugate the two fragments together (Figure 1). We selected an antibody CDR region from 806 humanized mAb with proven preclinical efficacy and clinical safety. Following the BAPTS procedure, one fragment with anti-EGFRvIII and another fragment with anti-CD3 mAb were constructed and expressed by CHO cells. Both fragments were purified by Protein L affinity chromatography (Figure 2a, left). Fragment A was composed of three peptides, CD3Lc, CD3Hc, and CD3-IntC. Similarly, fragment B was composed of two peptides, EGFRvIII-IntN and EGFRvIII-Lc (Figure 2a, right). A newly generated band was generated by SDS-PAGE analysis in the corresponding molecular weight, with a decreasing amount of antibody fragments as starting materials (Figure 2b). Since the reducing agent needed in the PTS reaction may interfere with correct folding and antibody structure, the proper reducing condition was screened for generating EGFRvIII-BsAb (Figure 2c). SDS-PAGE analysis showed that the reaction reached a plateau with the least fragment B residue under an optimal reaction condition of 0.5 mM TCEP at 37 ℃ for 2 h. Then, DHAA was used to neutralize extra reducing agent TCEP, as previously reported [26] (Figure 2c, left). The final product of EGFRvIII-BsAb was polished with an MMC ImpRes Multimodal Chromatography Column (Figure 2e), and was characterized with LC/MS and SPR. Ion mobility quadrupole time-of-flight (IMS Q-Tof) mass spectrometry was utilized to identify the molecular mass of the BsAb. After deglycosylation, the mass of the BsAb was 145,360 Da (Figure 2f). SEC-HPLC analysis showed a high purity of EGFRvIII-BsAb (Figure 2h), with a minimum amount of aggregation, half antibody, or hole–hole homodimer. EGFRvIII mAb was purified by Protein A affinity chromatography (Figure 2d).

### 3.2. In Vitro Activity of the EGFRvIII-BsAb

#### 3.2.1. Cellular Binding of the EGFRvIII-BsAb

The ability of EGFRvIII-BsAb to bind to target cells was verified by flow cytometry. We confirmed EGFRvIII-specific expression on the de2-7 mutant cell line U87MG.ΔEGFR (Figure 3a, left), but were not able to detect EGFRvIII on the surface of wild-type GBM cell line U87MG (Figure 3a, left). From the cellular binding experiment, the flow cytometry result showed that the EGFRvIII-BsAb specifically targeted EGFRvIII-positive cells without off-target binding to EGFRvIII-negative U87MG cells (Figure 3a, right), which indicated the superior specificity by the BsAb toward the EGFRvIII. The EGFRvIII-BsAb did bind with the CD3-positive Jurkat cell line, as well as the EGFRvIII-positive U87MG.ΔEGFR cell line, in a dose-dependent manner (Figure 3b). The affinities of EGFRvIII-BsAb with the antigens EGFRvIII and CD3 were determined by a Biacore assay (Figure 2g). It was previously reported that T-cell-dependent BsAbs with high affinity toward the CD3 antigen would lead to the biodistribution to T-cell-rich tissues and the fast elimination of the BsAbs [27]. Thus, we adapted the sequence of anti-CD3 antibody whose affinity to the CD3 was low, according to a previous report (US20130289127A1). The result confirmed that the EGFRvIII-BsAb had a lower affinity with the CD3 antigen compared with the parental CD3 mAb (1.95 × 10^-7^ M vs. 2.82 × 10^-8^ M), as designed. For binding to the EGFRvIII antigen, the BsAb appeared to have a lower affinity compared with the parental EGFRvIII mAb (1.60 × 10^-8^ M vs. 7.40 × 10^-10^ M, partly due to its monovalent structure.

#### 3.2.2. EGFRvIII-BsAb Redirected T Cells to Tumor Cells

To investigate the killing mechanism against the GBM cell mediated by BsAb, we evaluated the EGFRvIII-BsAb from the perspective of recruiting T cells to tumor cells. Under a light microscope, it was apparent that T cells were redirected to the tumor cell site in the presence of EGFRvIII-BsAb (Figure 3c). To further quantify recruitment ability, an FACS assessment was performed to evaluate the percentage of CD3- and EGFRvIII-positive cell clusters (Figure 3d). CD3-positive Jurkat cells were labeled with PKH26, and EGFRvIII-positive U87MG.ΔEGFR cells were labeled with CFSE. U87MG cells were labeled with CFSE as control. The result from the flow cytometry assessment demonstrated that the EGFRvIII-BsAb specifically redirected T cells to the EGFRvIII-positive cell line, and did not appear to affect the behavior of T cells against the EGFRvIII antigen-negative cell line U87MG (CFSE + /PKH26+ cell ratio was 20.86% vs. 2.05%). In contrast, no redirection ability of the EGFRvIII mAb was detected (CFSE+/PKH26+ cell ratio was 3.85%). This result indicated that the redirection of T cells was target-dependent.

#### 3.2.3. The EGFRvIII-BsAb Activated T Cells to Release Cytotoxic Cytokines

Next, we tested the activation of T cells by EGFRvIII-BsAb in vitro. The T-cell activation marker CD69 in CD4+ and CD8+ T cells was analyzed when target cells were incubated with the EGFRvIII-BsAb (Figure 4a). The CD4+CD69+ T cells in PBMCs were increased from 0.93% to 2.80%, and CD8+CD69+ T cells were increased from 0.18% to 3.94% with target cells only, showing that target cells had more influence on CD8+ T-cell activation. Interestingly, the CD4+ and CD69+ cells were increased from 0.93% to 19.33%, while the CD8+ and CD69+ positive cells were increased from 0.16% to 5.48% with EGFRvIII-BsAb only, indicating EGFRvIII-BsAb could activate both CD4+ and CD8+ T cells independently. In the presence of both target cells and the EGFRvIII-BsAb, the percentages of CD4+CD69+ and CD8+CD69+ cells reached 31.94% and 12.54%, respectively. These results demonstrated that CD4+ T-cell activation was mainly dependent on the EGFRvIII-BsAb, while CD8+ T-cell activation relied on the combined action of target cells and EGFRvIII-BsAb. Furthermore, this also supported the theory that CD8+ T cells were the main effector cells during the lymphocyte-mediated killing of tumors. To assess whether additional favorable cytokines were released in the process of T-cell activation by the EGFRvIII-BsAb, we measured the IFN-γ and IL-2 levels in the cell culture supernatant when effector cells were incubated with target cells and the EGFRvIII-BsAb or the EGFRvIII mAb. The cytokine release was barely detectable with the EGFRvIII mAb and the CD3 mAb, while the cytokines were remarkably increased with the EGFRvIII-BsAb (Figure 4b). A time- and antibody-concentration-dependent release of IFN-γ and IL-2 was also observed. In the earlier stage, IL-2 release rapidly reached a peak around 50 h after EGFRvIII-BsAb treatment. However, the IFN-γ release showed a continuous increase. In conclusion, EGFRvIII-BsAb did activate T cells, which led to cytokine release.

#### 3.2.4. EGFRvIII-BsAb Mediated T Cells to Lyse Tumor Cells In Vitro

To evaluate the cytotoxicity of the EGFRvIII-BsAb, we incubated tumor cells, effector cell PBMCs, and antibodies together, and then collected the cell culture to test the lactate dehydrogenase (LDH) release level (Figure 4c,d). Among all experimental groups, the EGFRvIII-BsAb showed potent cytotoxicity compared to its parental antibodies, EGFRvIII mAb and CD3 mAb, even at a low dosage of 0.01 ug/mL. Furthermore, the BsAb was superior in killing tumor cells compared with combo treatment with the EGFRvIII mAb and the CD3 mAb. This demonstrated that cytotoxicity was mediated through the synergistic effect by recruiting immune cells, rather than through a simple combined effect of the CD3 mAb and the EGFRvIII mAb (Figure 4c,d). In addition, the EGFRvIII-BsAb showed higher maximum killing activity and lower EC_50_ compared with parental EGFRvIII mAb (Figure 4d,e) in the dose–response cytotoxicity experiment. We also observed that the antitumor killing activity of the EGFRvIII mAb was mainly due to its Fc domain that binds to FcγRIII positive effector cells, including NK cells and macrophages (Figure 4c). Once ADCC and CDC function was depleted, no cytotoxicity of the EGFRvIII mAb was observed. This observation further proved that the killing ability of EGFRvIII-BsAb solely relied on its effect on T cells. This is highly advantageous for controllable T-cell activation without other immune effector cells being activated.

### 3.3. In Vivo Activity of the EGFRvIII-BsAb

#### 3.3.1. Tumor Growth Inhibition in Immunocompromised Mice by the EGFRvIII-BsAb

To evaluate the in vivo activity of the EGFRvIII-BsAb in NOD/SCID mice, we first explored the effect of the E/T ratio on the tumor growth. The U87MG.ΔEGFR cells expressing EGFRvIII, together with huPBMCs from a healthy donor at an E/T ratio of 1:1 and 2:1, were subcutaneously implanted into immunodeficient mice. The EGFRvIII-BsAb or placebo at 5 mg/kg were intraperitoneally administered every three days from the second day of tumor implantation (Figure 5a). Compared with the placebo group, the EGFRvIII-BsAb significantly inhibited U87MG.ΔEGFR-derived tumor growth. Under a high E/T ratio of 2:1, the average tumor volume was smaller, indicating that the existence of huPBMCs contributed to tumor inhibition. The most effective tumor inhibition was observed in the EGFRvIII-BsAb E/T 2:1 group. Further results demonstrated that complete tumor elimination was the synergetic effect of the EGFRvIII-BsAb and immune cells (Figure 5b–d). Then, in another subsequent in vivo experiment, an antitumor ability assessment of EGFRvIII-BsAb under a series of dosages (3 mg/kg, 1 mg/kg, 0.33 mg/kg) and EGFRvIII mAb (1.5 mg/kg) was carried out with PBS as a placebo (Figure 5e). From the result, the EGFRvIII-BsAb showed an initial inhibitory effect toward tumor growth at a concentration as low as 0.33 mg/kg, which was 5 times lower than the control EGFRvIII mAb (Figure 5f,g). At a high dose of 3 mg/kg, the EGFRvIII-BsAb was able to eliminate tumor cells completely, without an observed recurrence (Figure 5g). In addition, no body-weight decrease was observed among any of the groups. Interestingly, mice treated with the EGFRvIII mAb suffered from beard shedding, while this side effect was not observed in the groups treated with PBS buffer (placebo) or EGFRvIII-BsAb. To recap, EGFRvIII-BsAb showed promising therapeutic efficacy in vivo, with no obvious side effects observed.

#### 3.3.2. Pharmacokinetic (PK) Analysis

The PK parameters of EGFRvIII-BsAb and EGFRvIII mAb were evaluated in male BALB/c mice after a single intraperitoneal (i.p.) administration of 5 mg/kg. A noncompartment model was applied, and data showed that both EGFRvIII-BsAb and EGFRvIII mAb displayed a biphasic disposition (Figure 6). Interestingly, an long half-life was unexpectedly observed in the EGFRvIII-BsAb group (13.90 days) compared to the group that was administered the parental EGFRvIII mAb (3.56 days) (Figure 5h). The long circulation time of EGFRvIII-BsAb may be attributed to its lower affinity with the CD3 antigen, as previously documented [27]. This merit is especially advantageous in clinical treatments.

## 4. Discussion

Within the past 20 years, limited progress has been made in combatting GBM. Potential reasons include a highly heterogeneous brain tumor microenvironment, an impermeable BBB, and lack of T-cell infiltration. There have been many frustrating moments in the development of therapeutic antibodies and vaccines, including ABT-414 [28,29], an antibody–drug conjugate (ADC) against the EGFRvIII; Rindopepimut [30,31], a tumor vaccine against EGFRvIII; and mAb treatment in Phase III clinical trials [32]. According to disclosed clinical data, ABT-414 failed to show a survival benefit for patients compared with a placebo, with a severe corneal side effect caused by ADC’s on-target or off-target cytotoxicity [33]. Although several BsAbs against EGFRvIII have been developed, they have not progressed beyond preclinical studies [34,35]. Here, an IgG-like BsAb with low immunogenicity, excellent GBM tumor inhibition, and a long circulation half-life was reported to provide a potential approach for treating GBM. The high-quality material of the BsAb was generated through the BAPTS platform and multimodal BioProcess™ chromatography steps [36]. Compared with the most advanced hEGFRvIII-CD3 bi-scFv and TandAb [35,37], the IgG-like BsAb proved to have superior antitumor ability in vitro under a lower E/T ratio and EC_50_, and a much longer circulation time in vivo (13.90 days vs. 2.8 h [35]).

In our in vivo animal experiments, we used immunodeficient mice to imitate the immune-cell-infiltrating tumor microenvironment by reestablishing their immune systems with mixed huPBMCs and tumor cells, based on previous studies [28,37,38]. The tumor inhibitory ability of EGFRvIII-BsAb was verified. Human immune cells usually have a limited half-life in the murine background, which might limit the immune cells’ exertion when exhibiting full antitumor ability [19,39]. In the future, we will evaluate the efficacy and biodistribution of EGFRvIII-BsAb in an orthotopic tumor model with a humanized mouse model and/or a patient-derived GBM model to evaluate the full potential of EGFRvIII-BsAb. In addition, we observed beard shedding as a side effect in mice in the parental EGFRvIII mAb treatment group. This side effect may be due to the cross-reaction between EGFRvIII mAb and highly endogenously expressed EGF in skin, as previously reported [38]. However, no severe toxicities were observed in the EGFRvIII-BsAb treatment groups, even with a long circulation in vivo.

When we looked into the details of T-cell-dependent cytotoxicity mediated by EGFRvIII-BsAb, we found that the BsAb activated T cells and killed tumor cells in a dose-dependent manner. Moreover, according to the IFN-γ and IL-2 release assay, we depicted a cytokine-release surface diagram. The results indicated a time-dependent cytokine release in response to EGFRvIII-BsAb treatment, and implied that the immune function of the EGFRvIII-BsAb was controllable. Furthermore, a controlled cytokine release could assist in the design of a clinically safe starting dose, which would be based on a minimum anticipated biological effect level (“MABEL”) approach [40]. It is notable that EGFRvIII-BsAb mediated a potent killing effect against tumor cells at an E/T ratio as low as 5:1, and that it showed prolonged cytotoxicity against tumor cells compared with parental EGFRvIII mAb. As expected, once the ADCC and CDC functions were knocked out, EGFRvIII mAb was unable to inhibit tumor growth. EGFRvIII mAb’s antitumor efficacy largely relied on the Fc domain to bind with the FcγIIIR to kill cancer cells in the early stages, while EGFRvIII-BsAb activated T cells to proliferate and consistently present antitumor efficacy.

Solid tumors are characterized by the immune suppressive microenvironment, with the existence of myeloid-derived suppressor cells (MDSCs), regulatory T cells (Tregs), tumor-associated macrophages (TAM), and cancer-associated fibroblast (CAF), negatively hampering the killing of cytotoxic T cells [41]. However, the application of immune check point inhibitors recuperates tumor inhibition and changes a “cold tumor” into “hot tumor” [42,43]. It is noticeable that oncolytic virus and gene therapies have been widely tested in clinical trials for the treatment of GBM, with significant immune-cell infiltration and tolerance [44,45]. Preconditioning the tumor microenvironment with an oncolytic virus also proved to be an effective strategy for enhancing CD3-BsAb in immune-silent solid tumors [46]. Therefore, combination treatment consisting of immune checkpoint blockades or oncolytic virus and EGFRvIII-BsAb would show further applicable potential in the treatment of GBM.

## 5. Conclusions

We have demonstrated that EGFRvIII-BsAb redirected T cells to GBM cells and stimulated T cells to release cytotoxic cytokines to kill tumor cells. Both in vitro and in vivo experiments showed that EGFRvIII-BsAb was superior to the parental EGFRvIII mAb in antitumor efficacy and circulation time. This BsAb appeared as a potentially promising therapeutic reagent for GBM treatment in clinical applications.

## Figures and Tables

**Figure 1 biomedicines-09-00640-f001:**
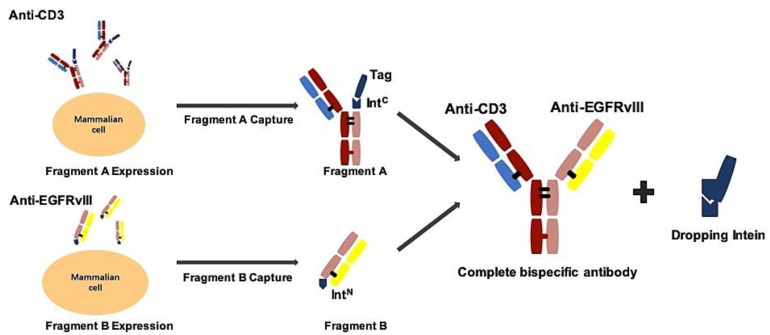
Schematic process of EGFRvIII-BsAb synthesis by the BAPTS platform. In this method, under 4–25 °C and reduced conditions, fragment A and fragment B were fused into a complete bispecific antibody, and the intein was dropped.

**Figure 2 biomedicines-09-00640-f002:**
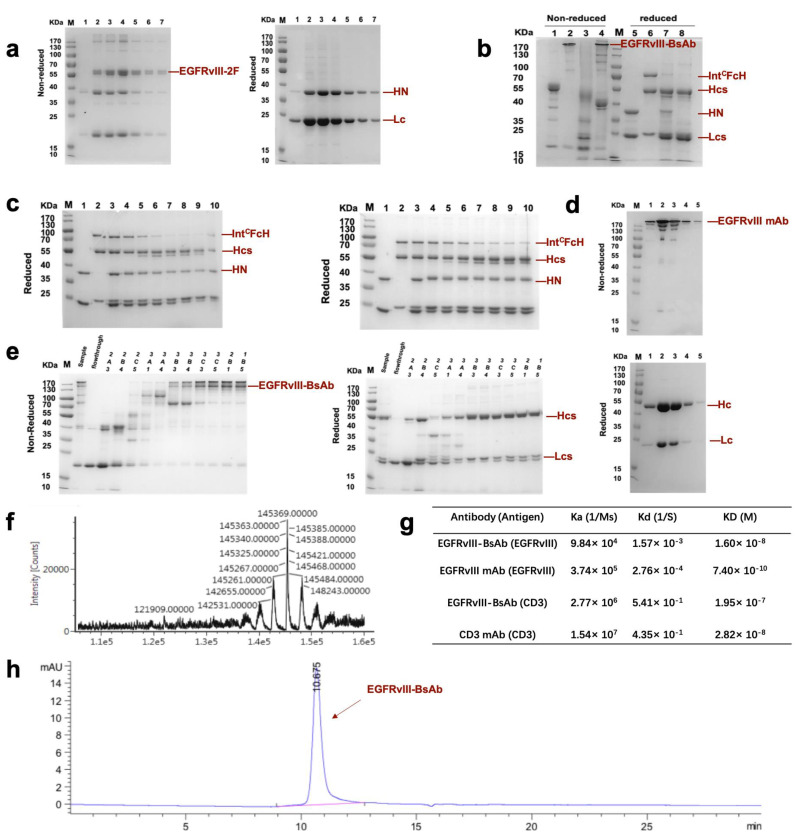
The generation and characterization of EGFRvIII-BsAb with the BAPTS method. (**a**) SDS-PAGE analysis of EGFRvIII-2F (fragment B) under nonreduced and reduced conditions. (**b**) SDS-PAGE analysis of BAPTS reaction. Bands 1,5—EGFRvIII-2F (fragment B); bands 2,6—CD3-3F (fragment A); bands 3,7—BAPTS reaction catalyzed by reduction agent; bands 4,8—complete bispecific antibody obtained by the BAPTS reaction, and followed by oxidization reagent reconstitution. (**c**) Reduced SDS-PAGE analysis of BAPTS reaction condition optimization. Bands 1–10 represent EGFRvIII-2F, CD3-3F, the mixture of 2F and 3F in the presence of 0 mM, 0.01 mM, 0.05 mM, 0.1 mM, 0.5 mM, 1 mM, 2 mM, and 5 mM reduction agent. (**d**) Nonreduced (upper) and reduced (lower) SDS-PAGE analysis of purification products of EGFRvIII mAb. (**e**) Nonreduced (left) and reduced (right) SDS-PAGE analysis of purification products of EGFRvIII-BsAb with an MMC ImpRes Multimodal Chromatography Column. (**f**) LC/MS analysis of deglycosylated nonreduced BsAb. (**g**) SPR assay of the dissociation constant of EGFRvIII-BsAb and its control mAbs against recombinant target antigens. (**h**) SEC-HPLC analysis of EGFRvIII-BsAb.

**Figure 3 biomedicines-09-00640-f003:**
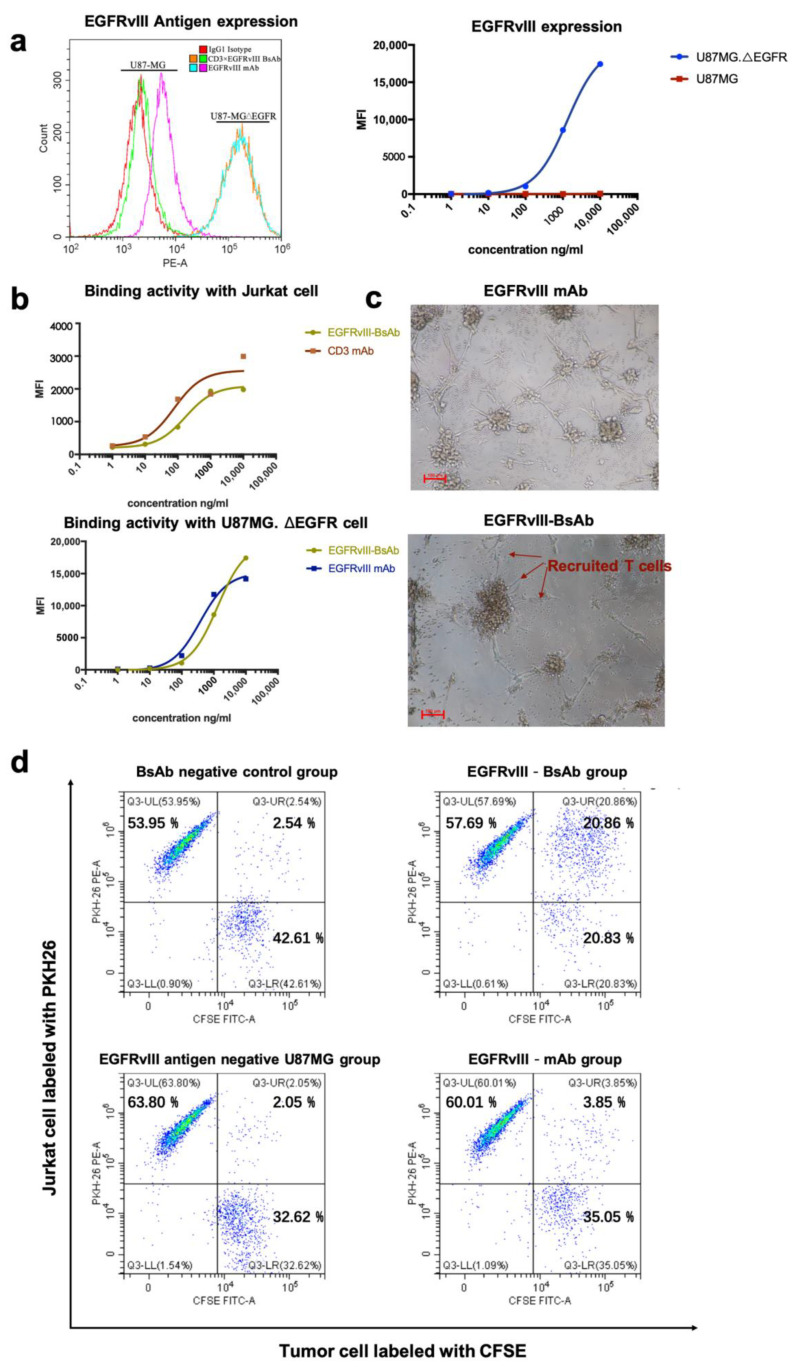
(**a**) EGFRvIII antigen expression level on mutant GBM cell line U87MG.ΔEGFR and wild-type GBM cell line U87MG. (**b**) A binding activity comparison of EGFRvIII-BsAb and CD3 mAb with Jurkat cells (CD3-positive) (upper), as well as a binding activity comparison of the EGFRvIII-BsAb and the EGFRvIII mAb with U87MG.ΔEGFR cells (EGFRvIII-positive) (lower). (**c**) Photographs of the redirection of T cells to cancer cells by 0.01 ng/mL EGFRvIII-BsAb or EGFRvIII mAb. (**d**) FACS analysis of the redirection of CD3+ Jurkat cells to cancer cells by EGFRvIII-BsAb. Jurkat (CD3+) cells labeled by PKH26 (PE-A), as well as U87MG.ΔEGFR cells labeled by CFSE (FITC-A).

**Figure 4 biomedicines-09-00640-f004:**
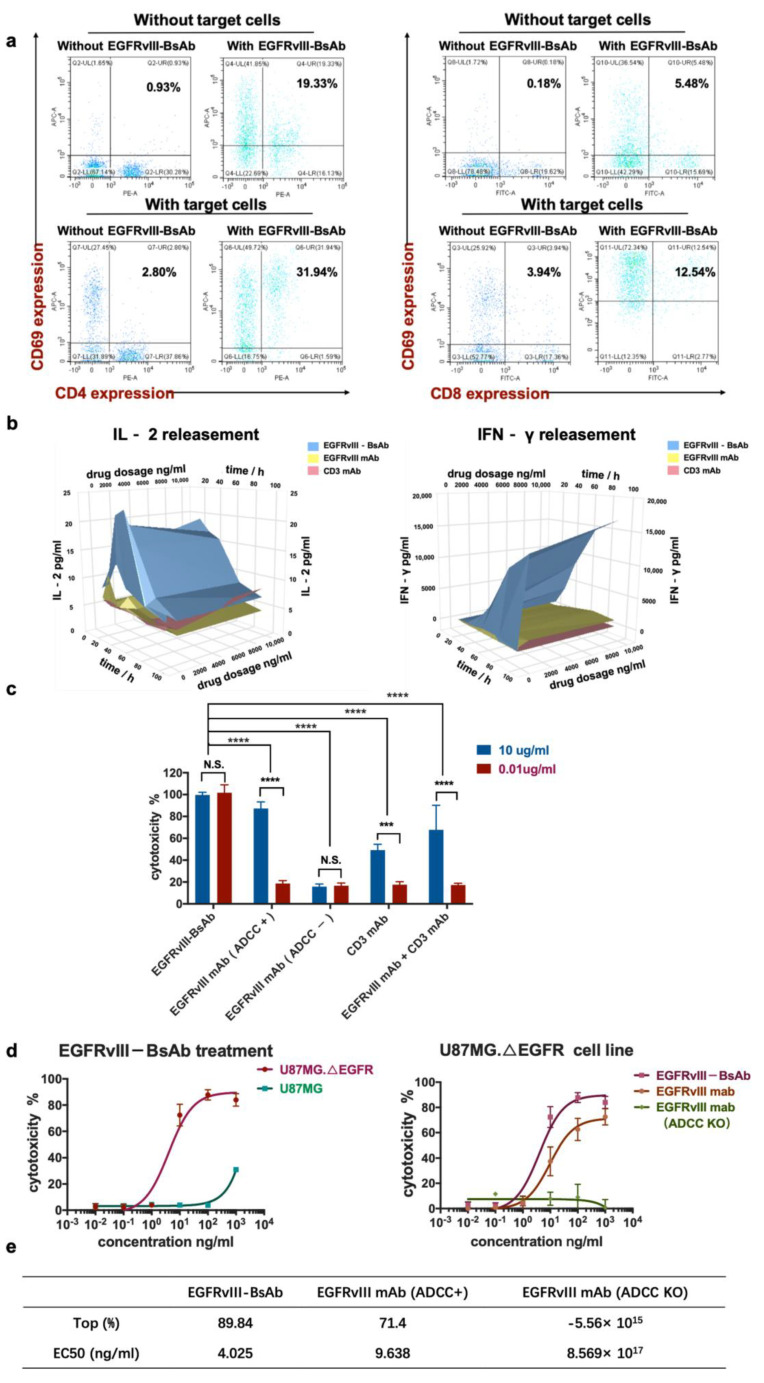
T-cell activation by EGFRvIII-BsAb. (**a**) CD69 expression on CD4+ and CD8+ T cells’ surface in response to target tumor cells or the EGFRvIII-BsAb. (**b**) The secreted levels of IFN-γ and IL-2 in culture supernatant detected after treatment with different EGFRvIII-BsAb concentrations by ELISA. (**c**) The cytotoxicity measurement of the EGFRvIII-BsAb and its parental antibodies or their combo treatment under low and middle antibody concentration by the LDH method (*p* < 0.0002 (***), *p* < 0.0001 (****)). (**d**) Dose-dependent antitumor activity evaluation of the EGFRvIII-BsAb on U87MG.ΔEGFR or U87MG (left). Dose-dependent antitumor activity evaluation of the EGFRvIII-BsAb compared with the EGFRvIII mAb and the CD3 mAb on U87MG.ΔEGFR cells (right). (**e**) Maximum cytotoxicity and EC_50_ comparison among EGFRvIII-BsAb and EGFRvIII mAb or ADCC-attenuated EGFRvIII mAb counterpart. Cytotoxicity measurement by LDH release assay.

**Figure 5 biomedicines-09-00640-f005:**
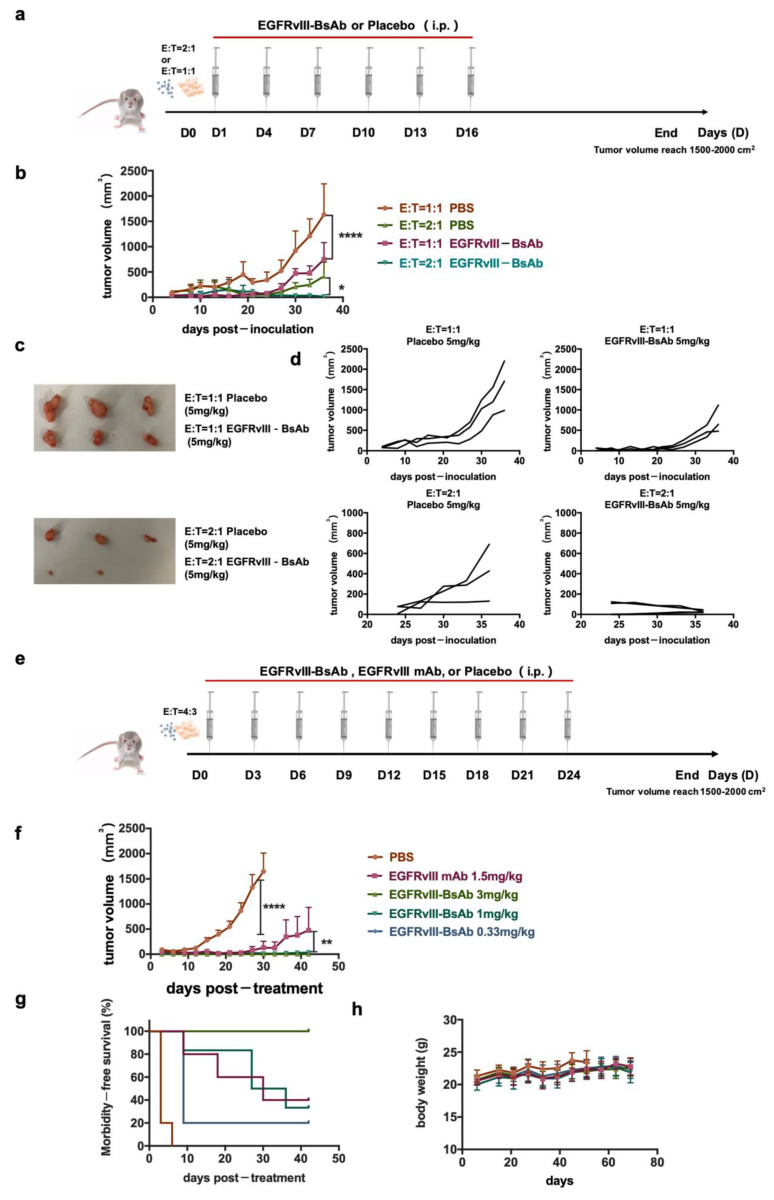
EGFRvIII-expression of tumor growth inhibition in vivo by EGFRvIII-BsAb. (**a**) Schematic schedule of tumor inoculation and treatment. (**b**) Time course of U87MG.ΔEGFR tumor-growing. Data are presented as mean ± SEM (*p* < 0.0332 (*), *p* < 0.0001 (****)). (**c**) Images of stripped tumors. (**d**) U87MG.ΔEGFR tumor-growing time course under various E/T ratios. Data are presented as measured tumor volumes from individual mice. (**e**) Schematic schedule of tumor inoculation and treatment. (**f**) Tumor-growing time course(*p* < 0.0021 (**), *p* < 0.0001 (****)). (**g**) Morbidity time course. (**h**) Body-weight time course.

**Figure 6 biomedicines-09-00640-f006:**
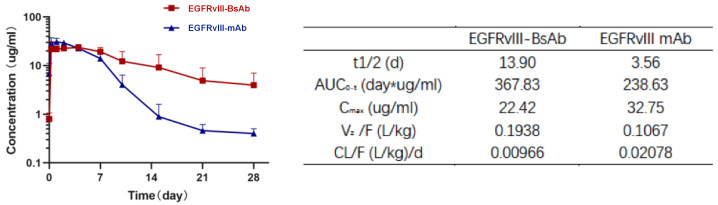
PK analysis of EGFRvIII-BsAb and EGFRvIII mAb in BALB/c mice. Antibody concentration of serum samples at different time points (**left figure**), and PK parameters analysis (**right table**).

## Data Availability

The data set supporting the conclusions of this article is included within the article.

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
