# Peer review of "A Rational Designed Novel Bispecific Antibody for the Treatment of GBM"

_biomedicines, 2021, doi:10.3390/biomedicines9060640_

Round 1
Reviewer 1 Report
Dear Sirs. With great enthusiasm I walked through your manuscript and it sounds very interesting. It seems to me that you have a wonderful U87 model for in vitro experiment but, unfortunately, GBM is heterogeneous and testing your proof of concept will be required primary tissue cultures. Especially it is important since adherent cells can not recapitulate the aggressive behavior of isolated from patient glioma cells. Secondly, to grow the tumor it requires to have a special environment, brain environment. Your sc model is artificial and can not represent true GBM scenario. Overall, you have a good start but in vitro and in vivo data, in my opinion, need to be reassessed.
Reviewer 2 Report
The manuscript by Sun et al. presents an anti-glioblastoma effect of EGFRvIII Bispecific Antibody. Clearly, this study is well designed and the manuscript shows novel data.
The main concerns are following:
- The language should be improved.
- The abbreviated form should be used in subsequent repetitions (e.g. GBM).
- It would be reasonable to increase font size in some legends, particularly in Fig. 4b, Fig. 4d and Fig. 5.
- Fig. 4c: it seems that there is a statistical significance between 10 ug/ml and 0.01 ug/ml in EGFRvIII mAb (ADCC+), CD3 mAb, and EGFRvIII mAb + CD3 mAb groups. If so, the significance should marked in the graph.
Author Response
Point 1: The language should be improved.
Responses 1: We appreciate review’s comment on: “The language should be improved”. We have completed thorough language editing for entire manuscript. Detailed modifications have been attached to a modified version of the original manuscript with tracking all the changes.
Point 2: The abbreviated form should be used in subsequent repetitions (e.g. GBM).
Response 2: Thanks for your suggestion! We’ve adopted the abbreviated form such as “GBM” in subsequent repetitions.
Point 3: It would be reasonable to increase font size in some legends, particularly in Fig. 4b, Fig. 4d and Fig. 5.
Response 3: We increased font size and made the font bold if necessary in Fig. 4b, Fig 4d and Fig. 5. Besides, we split the Fig.5 into two pictures to improve the quality and resolution of the Figure.
Point 4: Fig. 4c: it seems that there is a statistical significance between 10 ug/ml and 0.01 ug/ml in EGFRvIII mAb (ADCC+), CD3 mAb, and EGFRvIII mAb + CD3 mAb groups. If so, the significance should marked in the graph.
Response 4: Yes, there is a statistical significance between 10ug/ml and 0.01ug/ml in EGFRvIII mAb (ADCC+), CD3 mAb, and EGFRvIII mAb + CD3 mAb groups. Thanks for pointing out the flaw in our data process. Your kind advice has been well taken and the significances have been marked between different groups in the revision.
Reviewer 3 Report
in this paper, Sun et al describe the methodology, the biochemical characterization and the activity of bispecific antibodies targeting EGFRvIII for the treatment of glioblastoma. the paper is well written and easily comprehensible for non expert readers. however, it should be revised by a english mothertongue writer since some archaic words are used.
Author Response
Point 1: The paper is well written and easily comprehensible for non expert readers. however, it should be revised by an English mothertongue writer since some archaic words are used.
Responses 1: Thanks for your comments enabling us to improve the manuscript! Your suggestions have been well taken. The manuscript has been edited by the language service provided by MDPI. Detailed modifications have been attached to this response and a modified version of the original manuscript with tracking all the changes has been uploaded.

Round 2
Reviewer 1 Report
Dear Sirs, Although you provided significant justification for use of sc GBM model, Im still standing my ground and considering your data is artificial, regardless of FDA or any other studies. At the same time, I think your data is solid, can be a " food for thought" at this point and , hopefully, you will validate it using ic GBM model one day.